# Half-quantum vortices and walls bounded by strings in the polar-distorted phases of topological superfluid $^3$He

J. T. Mäkinen [1], V. V. Dmitriev[2], J. Nissinen[1], J. Rysti[1], G. E. Volovik[1,3], A. N. Yudin[2], K. Zhang[1,4] & V. B. Eltsov[1]

Symmetries of the physical world have guided formulation of fundamental laws, including relativistic quantum field theory and understanding of possible states of matter. Topological defects (TDs) often control the universal behavior of macroscopic quantum systems, while topology and broken symmetries determine allowed TDs. Taking advantage of the symmetry-breaking patterns in the phase diagram of nanoconfined superfluid $^3$He, we show that half-quantum vortices (HQVs)—linear topological defects carrying half quantum of circulation—survive transitions from the polar phase to other superfluid phases with polar distortion. In the polar-distorted A phase, HQV cores in 2D systems should harbor non-Abelian Majorana modes. In the polar-distorted B phase, HQVs form composite defects—walls bounded by strings hypothesized decades ago in cosmology. Our experiments establish the superfluid phases of $^3$He in nanostructured confinement as a promising topological media for further investigations ranging from topological quantum computing to cosmology and grand unification scenarios.

[1] Low Temperature Laboratory, Department of Applied Physics, Aalto University, FI-00076 Aalto, Finland. [2] P. L. Kapitza Institute for Physical Problems of RAS, Moscow, Russian Federation 119334. [3] Landau Institute for Theoretical Physics, Chernogolovka, Russian Federation 142432. [4] Department of Mathematics and Statistics, University of Helsinki, P.O. Box 68FI-00014 Helsinki, Finland. Correspondence and requests for materials should be addressed to J.T.M. (email: jere.makinen@aalto.fi)

Topological defects generally form in any symmetry-breaking phase transitions. The exact nature of the resulting TDs depends on the symmetries before and after the transition. Our universe has undergone several such phase transitions after the Big Bang. As a consequence, a variety of TDs might have formed during the early evolution of the Universe, where phase transitions lead to unavoidable defect formation via the Kibble–Zurek mechanism[1,2]. Experimentally accessible energy scales $\lesssim 1$ TeV are currently limited to times $t \gtrsim 10^{-12}$ s after the Big Bang by the Large Hadron Collider. Theoretical understanding may be extended up to the Grand Unification energy scales $\lesssim 10^{15}$ GeV of the electroweak and strong forces ($t \gtrsim 10^{-36} \ldots 10^{-32}$ s). The nature of the interactions before this epoch remains unknown[3,4], but yet unobserved cosmic TDs, the nature of which depends on the Grand Unified Theory (GUT) in question, may help us limit the possibilities. Predictions exist for point defects, such as the t'Hooft–Polyakov magnetic monopole[5,6], linear defects or strings[1], surface defects or domain walls[7], and three-dimensional textures[8].

Even though cosmic TDs have not been detected, many of their condensed-matter analogs have been reproduced in the laboratory, where they have an enormous impact on the behavior of the materials they reside in[9]. Examples include vortices in super-conductors[10], vortices and monopoles in ultracold gases[11,12], and skyrmions in chiral magnets[13]. Superfluid phases of $^3$He offer an experimentally accessible system to study a variety of TDs and the consequences of symmetry-breaking patterns owing to its rich order-parameter structure resulting from the $p$-wave pairing. Analogs of exotic TDs, such as the Witten string[14]—the broken-symmetry-core vortex in superfluid $^3$He-B[15–17], the skyrmion texture in superfluid $^3$He-A[18], and the Alice string[19]—half-quantum vortex (HQV) in the polar phase of superfluid $^3$He[20], have been observed.

Of particular interest are composite defects—combinations of TDs and/or non-topological defects of different dimensionality[21–23]. Such defects appear in some GUTs and even in the Standard Model, where the Nambu monopole may terminate an electroweak string[24,25]. There are two mechanisms for the formation of composite defects: the hierarchy of energy/interaction length scales[23,26,27], and the hierarchy (sequential order) of the symmetry-breaking phase transitions[22,28]. Composite defects originating from the hierarchy of length scales of condensation, magnetic, and spin-orbit energies are well-known in superfluid $^3$He. For example, the spin-mass vortex in $^3$He-B[23,29] has a hard core of the coherence-length size, defined by the condensation energy, and a soliton tail with thickness of the much larger spin-orbit length. A HQV originally predicted to exist in the chiral superfluid $^3$He-A[30] has a similar structure with the soliton tail, which makes these objects energetically unfavorable.

Composite defects related to the hierarchy of symmetry-breaking phase transitions were discussed in the context of the GUT scenarios by Kibble, Lazarides, and Shafi[22,28]. Here the GUT symmetry, such as $Spin(10)$, is broken into the Pati–Salam group $SU(4) \times SU(2) \times SU(2)$, which in turn is broken to the Standard Model symmetry group $SU(3) \times SU(2) \times U(1)$. At the first transition, the linear defects—cosmic strings—become topologically stable, while after the second transition they are no longer supported by topology and form the boundaries of the non-topological domain walls, henceforth referred to as Kibble–Lazarides–Shafi (KLS) walls. To the best of our knowledge, observations of KLS walls bounded by strings have not been reported previously.

In this work, we explore experimentally the composite defects formed by both the hierarchy of energy scales and the hierarchy of symmetry-breaking phase transitions allowed by the phase diagram of superfluid $^3$He confined in nematically ordered aerogel-like material called nafen. In our sample, a sequence of the polar, chiral polar-distorted A (PdA) and fully gapped polar-distorted B (PdB) phases occurs on cooling from the normal state[31], see Fig. 1c. Previously, we established a procedure to form topologically protected HQVs in the polar phase[20]. At the transition from the polar phase to the PdA phase, we expect the HQVs to acquire spin-soliton tails with the width of the spin-orbit length, which is much larger than the coherence-length size of vortex cores. On a subsequent transition to the PdB phase, the symmetry breaks in such a way that HQVs lose topological protection and may exist only as boundaries of the non-topological KLS walls. Simultaneously, the spin solitons between HQVs are preserved in the PdB phase and such an object becomes a doubly-composite defect. Naively, however, one would expect that a much stronger tension of the KLS wall compared to that of the spin soliton, would lead to collapse of an HQV pair, possibly to a singly quantized vortex with an asymmetric core[15–17,32,33].

Here we report evidence that HQVs do exist in the superfluid PdA and PdB phases of $^3$He. We create an array of HQVs by rotating the container with the angular velocity $\Omega$ in zero magnetic field during the transition from the normal fluid to the polar phase[20] and proceed by cooling the sample through consecutive transitions to the PdA and PdB phases. The HQVs are identified based on their nuclear magnetic resonance (NMR) signature as a function of temperature and $\Omega$. A characteristic satellite peak present in the NMR spectrum confirms that the HQVs survive in the PdA phase, where they provide experimental access to vortex-core-bound Majorana states[34,35]. Moreover, the HQVs are found to survive the transition to the PdB phase. The observed features of the NMR spectrum in the PdB phase suggest that a KLS wall emerges between a pair of HQVs already connected by the spin soliton. Evidently, the tension of the KLS wall is not sufficient to overcome the pinning of HQVs in nafen. Vortex pinning allows us to study the properties of the out-of-equilibrium vortex state created during the superfluid phase transitions while suppressing the vortex dynamics. Simultaneously pinning does not affect the symmetry-breaking pattern leading to formation of the KLS walls. Our results show that pinned TDs, once created, may be transferred to new phases of matter with engineered topology[36–38].

## Results

**Half-quantum vortices in the PdA phase.** The superfluid phase diagram under confinement by nafen[31]—a nanostructured material consisting of nearly parallel strands made of $Al_2O_3$, c.f. Fig. 1b—differs from that of the bulk $^3$He; the critical temperature is suppressed and, more importantly, new superfluid phases—the polar, PdA, and PdB phases—are observed. We refer to the Supplementary Note 1 for a detailed discussion on these phases and their symmetries and focus on our observations regarding the HQVs in the PdA and PdB phases.

The order parameter of the PdA phase can be written as

$$A_{\alpha j} = \sqrt{\frac{1+b^2}{3}}\, \Delta_{\mathrm{PdA}}\, e^{i\phi}\, \hat{\mathbf{d}}_\alpha \left( \hat{\mathbf{m}}_j + ib\hat{\mathbf{n}}_j \right), \tag{1}$$

where the orbital anisotropy vectors $\hat{\mathbf{m}}$ and $\hat{\mathbf{n}}$ form an orthogonal triad with the Cooper pair orbital angular momentum axis $\hat{\mathbf{l}} = \hat{\mathbf{m}} \times \hat{\mathbf{n}}$, and $\hat{\mathbf{d}}$ is the spin anisotropy vector. Vector $\hat{\mathbf{m}}$ is fixed parallel to the nafen strands. The amount of polar distortion is characterized by a dimensionless parameter $0 < b < 1$ and $\Delta_{\mathrm{PdA}}(T, b)$ is the maximum gap in the PdA phase. The order parameter of the polar phase is obtained for $b = 0$, while $b = 1$ produces the order parameter of the conventional A phase.

In our experiments, we use continuous-wave NMR techniques to probe the sample, see Methods for further details. In the

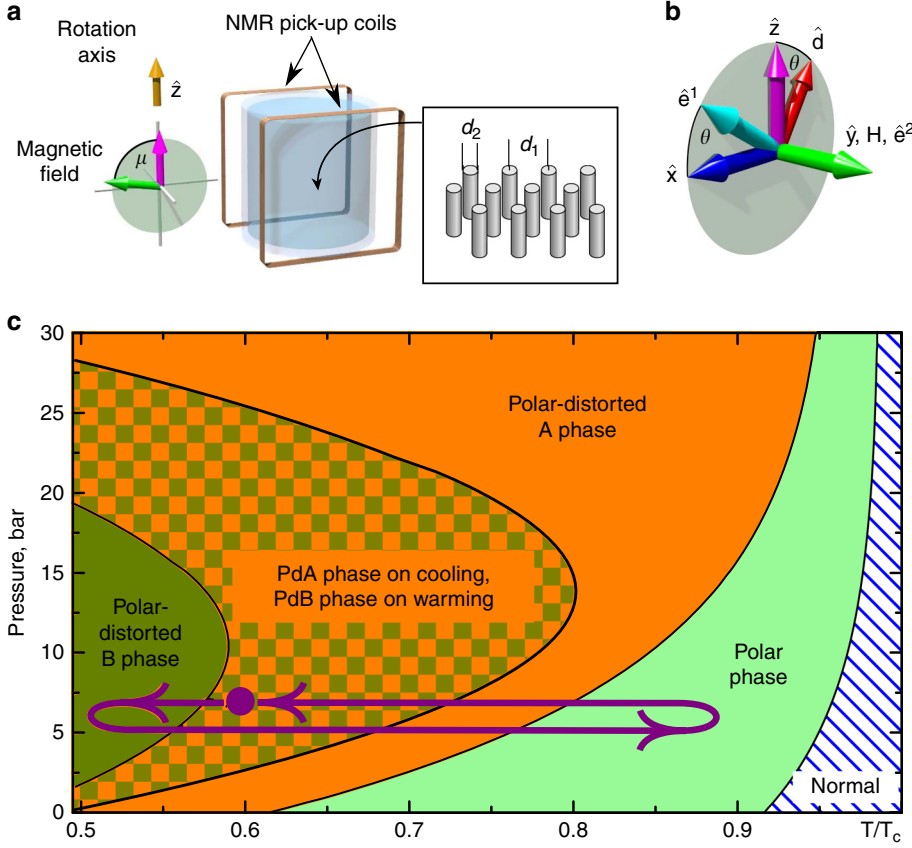

**Fig. 1** The experimental setup and superfluid phase diagram in nanoconfinement. **a** The $^3$He sample is confined within a cylindrical container filled with commercially available nanomaterial called nafen-90 (where the number refers to its density in mg cm$^{-3}$) with uniaxial anisotropy, which consists of nearly parallel Al$_2$O$_3$ strands with $d_2 \approx 8$ nm diameter, separated by $d_1 \approx 50$ nm on average. The strands are oriented predominantly along the axis denoted as $\hat{z}$. The sample can be rotated with angular velocities up to 3 rad s$^{-1}$ around the same axis $\hat{z}$. The sample is surrounded by rectangular nuclear magnetic resonance (NMR) pick-up coils. The static magnetic field transverse to the NMR coils can be oriented at an arbitrary angle $\mu$ with respect to the $\hat{z}$ axis. **b** The magnetic field, oriented along the $y$-direction ($\mu = \pi/2$) in this figure, locks the $\hat{e}^2$-vector in the polar-distorted B phase order parameter, Eq. (4). Vectors $\hat{d}$ and $\hat{e}^1$ are free to rotate in the $xz$-plane by angle $\theta$. **c** Sketch of the superfluid phase diagram in our sample in units of $T_c$ of the bulk fluid[31]. The purple arrows illustrate the thermal cycling used in the measurements and the purple marker shows a typical measurement point within the region where either polar-distorted phase can exist, depending on the direction of the temperature sweep. The thermal cycling is performed at constant 7 bar pressure

superfluid state, the spin-orbit coupling provides a torque acting on the precessing magnetization, which leads to a shift of the resonance from the Larmor value $\omega_L = |\gamma|H$, where $\gamma = -2.04 \times 10^8$ s$^{-1}$ T$^{-1}$ is the gyromagnetic ratio of $^3$He. The transverse resonance frequency of the bulk fluid with magnetic field in the direction parallel to the strand orientation, i.e. $\mu = 0$ in Fig. 1a, is[31]

$$\Delta\omega_{PdA} = \omega_{PdA} - \omega_L \approx \frac{\Omega_{PdA}^2}{2\omega_L}, \qquad (2)$$

where $\Omega_{PdA}$ is the frequency of the longitudinal resonance in the PdA phase at $\mu = \pi/2$. The NMR line retains its shape during the second-order phase transition from the polar phase but renormalizes the longitudinal resonance frequency due to appearance of the order-parameter component with $b$.

Quantized vortices are linear topological defects in the order-parameter field carrying non-zero circulation. In the PdA phase, quantized vortices involve phase winding by $\phi \rightarrow \phi + 2\pi\nu$ and possibly some winding of the $\hat{d}$ vector. The typical singly quantized vortices, also known as phase vortices, have $\nu = 1$ and no winding of the $\hat{d}$-vector, while the HQVs have $\nu = \frac{1}{2}$ and winding of the $\hat{d}$-vector by $\pi$ on a loop around the HQV core so

that sign changes of $\hat{d}$ and of the phase factor $e^{i\phi}$ compensate each other. The reorientation of the $\hat{d}$-vector leads to the formation of $\hat{d}$-solitons—spin-solitons connecting pairs of HQVs. The soft cores of the $\hat{d}$-solitons provide trapping potential for standing spin waves[39].

Since the $\hat{m}$-vector is fixed by nafen parallel to the anisotropy axis, the $\hat{l}$-vector lies on the plane perpendicular to it, prohibiting the formation of continuous vorticity[40] like the double-quantum vortex in $^3$He-A[41]. Some planar structures in the $\hat{l}$-vector field, such as domain walls[42] or disclinations, remain possible but the effect of the $\hat{l}$-texture on the trapping potential for spin waves is negligible due to the large polar distortion[31] (i.e. for $b \ll 1$). Recent theoretical work[43] provides arguments why formation of HQVs in the polar phase is preferred compared to the undistorted A phase. Studying whether HQVs are formed in the transition from the normal phase to the PdA phase with finite polar distortion ($0 < b < 1$) remains a task for the future. In our case, the PdA phase is obtained via the second-order phase transition from the polar phase with preformed HQVs. We already know[20] that the maximum tension from the spin-soliton in the polar phase (for $\mu = \pi/2$) is insufficient to overcome HQV pinning. Thus, survival of HQVs in the PdA phase is expected.

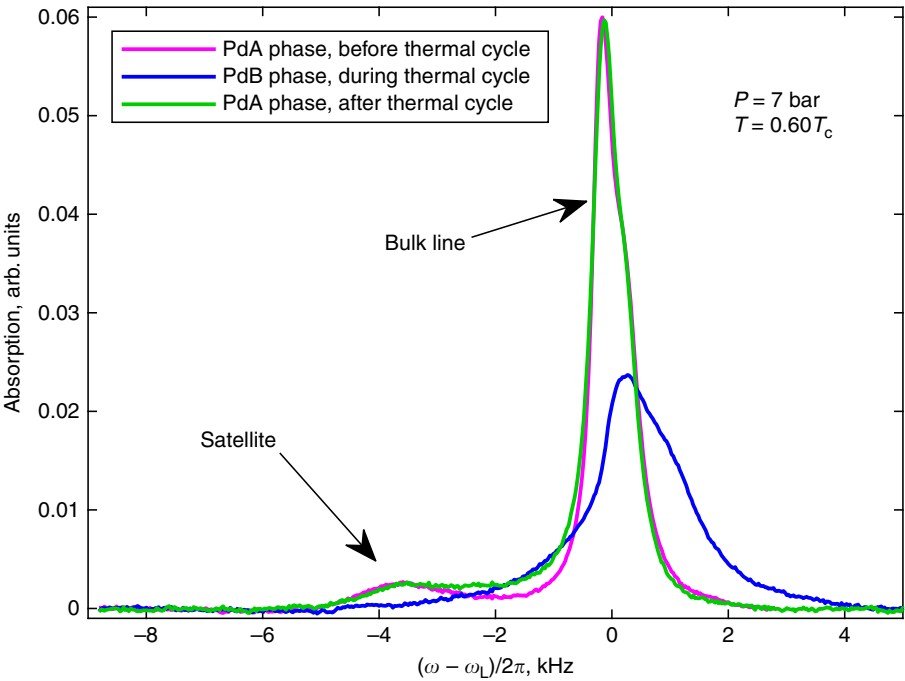

**Fig. 2** Survival of HQVs during phase transitions. The plot shows the measured NMR spectra in transverse ($\mu = \pi/2$) magnetic field in the presence of HQVs. HQVs were created by rotation with 2.5 rad s$^{-1}$ during the transition from normal phase to the polar phase. The NMR spectrum includes the response of the bulk liquid and the $\hat{\mathbf{d}}$-solitons, which appear as a characteristic satellite peak at lower frequency. The satellite intensity in the PdA phase remains unchanged after thermal cycling presented in Fig. 1c. The NMR spectrum in the PdB phase at the same temperature, measured between the two measurements in the PdA phase, is shown for reference

Moreover, we note that even for $|b| = 1$ and in the absence of pinning, a pair of HQVs, once created, should remain stable with finite equilibrium distance corresponding to cancellation of vortex repulsion and tension from the soliton tail[18].

In the presence of HQVs, the excitation of standing spin waves localized on the soliton leads to a characteristic NMR satellite peak in transverse ($\mu = \pi/2$) magnetic field, c.f. Fig. 2, with frequency shift

$$\Delta\omega_{\text{PdAsat}} = \omega_{\text{PdAsat}} - \omega_{\text{L}} \approx \lambda_{\text{PdA}} \frac{\Omega_{\text{PdA}}^2}{2\omega_{\text{L}}}, \qquad (3)$$

where $\lambda_{\text{PdA}}$ is a dimensionless parameter dependent on the spatial profile (texture) of the order parameter across the soliton. For an infinite 1D $\hat{\mathbf{d}}$-soliton, one has $\lambda_{\text{PdA}} = -1$, corresponding to the zero-mode of the soliton[18,20,44]. The measurements in the supercooled PdA phase, Fig. 3a, at temperatures close to the transition to the PdB phase give value $\lambda_{\text{PdA}} \approx -0.9$, which is in good agreement with theoretical predictions and earlier measurements in the polar phase with a different sample[20]. This confirms that the structure of the $\hat{\mathbf{d}}$-solitons connecting the HQVs is similar in polar and PdA phases and the effect of the orbital part to the trapping potential can safely be neglected. Detailed analysis of the satellite frequency shift as a function of magnetic field direction in the PdA phase remains a task for the future.

**Half-quantum vortices in the PdB phase.** Since the HQVs are found both in the polar and PdA phases, it is natural to ask what is their fate in the PdB phase? The number of HQVs in the polar and PdA phases can be estimated from the intensity (integrated area) of the NMR satellite, a direct measure of the total volume occupied by the $\hat{\mathbf{d}}$-solitons[20]. When cooling down to the PdB phase from the PdA phase, one naively expects the HQVs and the related NMR satellite to disappear since isolated HQVs cease to be protected by topology in the PdB phase. However, the

measured satellite intensity in the PdA phase before and after visiting the PdB phase remained unchanged, c.f. Fig. 2, which is a strong evidence in favor of the survival of HQVs in the phase transition to the PdB phase. Theoretically, it is possible that HQVs survive in the PdB phase as pairs connected by domain walls, i.e., as walls bounded by strings[22]. For very short separation between HQVs in a pair and ignoring the order-parameter distortion by confinement, such construction may resemble the broken-symmetry-core single-quantum vortex of the B phase[16]. In our case, however, the HQV separation in a pair exceeds the core size by 3 orders of magnitude. Let us now consider this composite defect in more detail.

The order parameter of the PdB phase can be written as

$$A_{\alpha j} = \sqrt{\frac{1 + 2q^2}{3}} \Delta_{\text{PdB}} e^{i\phi} (\hat{\mathbf{d}}_\alpha \hat{\mathbf{z}}_j + q_1 \hat{\mathbf{e}}_\alpha^1 \hat{\mathbf{x}}_j + q_2 \hat{\mathbf{e}}_\alpha^2 \hat{\mathbf{y}}_j), \qquad (4)$$

where $|q_1|, |q_2| \in (0, 1)$, $|q_1| = |q_2| \equiv q$ describes the relative gap size in the plane perpendicular to the nafen strands, $\hat{\mathbf{e}}^1$ and $\hat{\mathbf{e}}^2$ are unit vectors in spin-space forming an orthogonal triad with $\hat{\mathbf{d}}$, and $\Delta_{\text{PdB}}(T, q)$ is the maximum gap in the PdB phase. For $q = 0$, one obtains the order parameter of the polar phase, while $q = 1$ recovers the order parameter of the isotropic B phase. We extract the value for the distortion factor, $q \sim 0.15$ at the lowest temperatures from the NMR spectra using the method described in ref. [45], see Supplementary Note 6 for the measurements of $q$ in the full temperature range.

In transverse magnetic field **H** exceeding the dipolar field, the vector $\hat{\mathbf{e}}^2$ becomes locked along the field, while vectors $\hat{\mathbf{d}}$ and $\hat{\mathbf{e}}^1$ are free to rotate around the axis $\hat{\mathbf{y}}$, directed along **H**, with the angle $\theta$ between $\hat{\mathbf{d}}$ and $\hat{\mathbf{z}}$, c.f. Fig. 1b. The order parameter of the PdB phase in the vicinity of an HQV pair has the following properties. The phase $\phi$ around the HQV core changes by $\pi$ and the angle $\theta$ (and thus vectors $\hat{\mathbf{d}}$ and $\hat{\mathbf{e}}^1$) winds by $\pi$. Consequently, there is a phase jump $\phi \to \phi + \pi$ and related sign flips of vectors

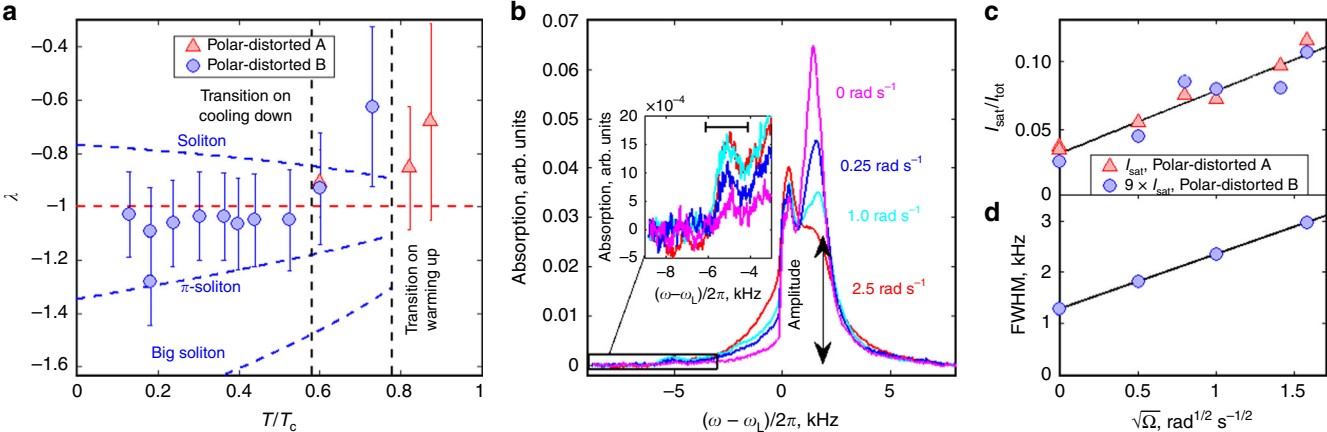

**Fig. 3** NMR spectra and spin-solitons in the polar-distorted phases. **a** Frequency shift of a characteristic satellite peak in the NMR spectrum expressed via parameter $\lambda$ as a function of temperature in the PdA and PdB phases. In the PdA phase, the measured values reside slightly above the theoretical prediction for a $\hat{\mathbf{d}}$-soliton with $\pi$ winding, shown as the red dashed line. The difference is believed to be caused by disorder introduced by nafen, as in the polar phase[20,70]. The corresponding values in the PdB phase for the lowest-energy $\hat{\mathbf{d}}$-soliton (marked "soliton") and its antisoliton (marked "big soliton"), as well as the combined $\pi$-soliton (see text) are shown as dashed blue lines. The $\pi$-soliton values turn out to be in the same ratio with respect to the experimental points as in the PdA phase. The error bars are based on the spectral width of the observed feature and denote the uncertainty in the position of the satellite peak, as illustrated by the black bar in the inset of **b** for the PdB phase. The range marked by an error bar corresponds to the doubled full width at half maximum (FWHM) of the satellite peak in the PdB phase (1 kHz). In the PdA phase, due to improved signal-to-noise ratio, the uncertainty is twice smaller (0.5 kHz). **b** The plot shows the measured NMR spectrum in the PdB phase at 0.38 $T_c$ for different HQV densities, controlled by the angular velocity $\Omega$ at the time of crossing the $T_c$. The presence of KLS walls produces characteristic features seen both as widening of the main line (with small positive frequency shift) and as a satellite peak with a characteristic negative frequency shift. The inset shows magnified view of the satellite peak. **c** The satellite intensity in the PdA phase at 0.60 $T_c$ (blue circles) and in the PdB phase multiplied by a factor of 9 (red triangles) at 0.38 $T_c$ show the expected $\sqrt{\Omega}$-scaling. The solid black line is a linear fit to the measurements including data from both phases. The non-zero $\Omega = 0$ intersection corresponds to vortices created by the Kibble–Zurek mechanism[1,2,20]. **d** The FWHM of the main line, determined from the spectrum in **b**, gives FWHM $\approx$3 kHz for 2.5 rad s$^{-1}$. FWHM for other angular velocities is recalculated from the amplitude of the main NMR line, shown in **b**, assuming constant area

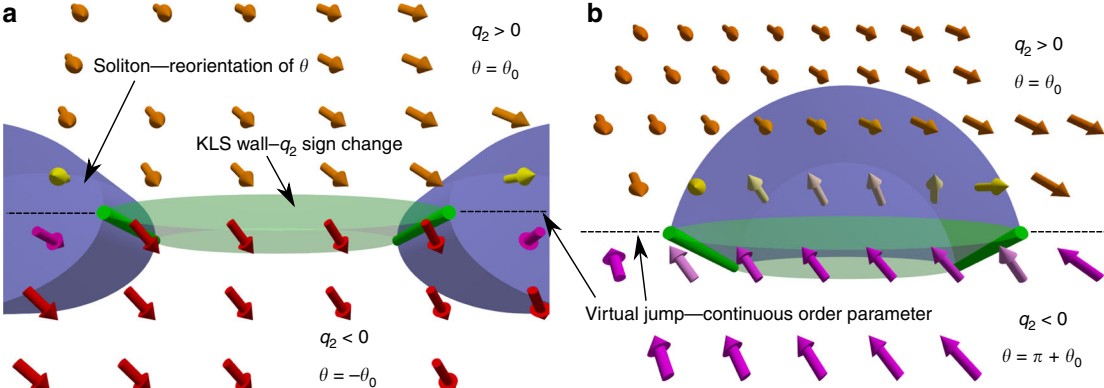

**Fig. 4** Kibble–Lazarides–Shafi (KLS) wall configurations in the PdB phase. Each HQV core terminates one soliton—reorientation of the spin part of the order parameter denoted by the angle $\theta$—and one KLS wall. The orientation of the $\hat{\mathbf{d}}$-vector is shown as arrows where their color indicates the angle $\theta$, based on numerical calculations (Supplementary Figure 2). **a** The KLS wall is bound between a different pair of HQV cores as the soliton. Ignoring the virtual jumps, the angle $\theta$ winds by $\pi - 2\theta_0$ across the soliton and by $2\theta_0$ across the KLS wall. The order parameter is continuous across the virtual jumps, where $\phi \rightarrow \phi + \pi$, $\theta \rightarrow \theta + \pi$, and $q_2 \rightarrow -q_2$. **b** The soliton and the KLS wall are bound between the same pair of HQV cores. The total winding of the $\hat{\mathbf{d}}$-vector is $\pi$ across the structure. In principle, the KLS wall may lie inside or outside the soliton. Here the KLS wall and the soliton are spatially separated for clarity

$\hat{\mathbf{d}}$ and $\hat{\mathbf{e}}^1$ along some direction in the plane perpendicular to the HQV core. In the presence of order-parameter components with $q > 0$, Eq. (4) remains single-valued if, and only if, $q_2$ also changes sign. We conclude that the resulting domain wall separates the degenerate states with $q_2 = \pm q$ and together with the bounding HQVs has a structure identical to the domain wall bounded by strings—the KLS wall—proposed by Kibble, Lazarides, and Shafi in refs. [22,28].

The KLS wall and the topological soliton have distinct defining length scales[17,33]—the KLS wall has a hard core of the order of

$\xi_W \equiv q^{-1}\xi$, where $\xi$ is the coherence length, and the soliton has a soft core of the size of the dipole length $\xi_D \gg \xi_W$. The combination of these two objects may emerge in two different configurations illustrated in Fig. 4. The minimization of the free energy (Supplementary Notes 2 and 3) shows that in the PdB phase, the lowest-energy spin-soliton corresponds to winding of the $\hat{\mathbf{d}}$-vector by $\pi - 2\theta_0$, where $\sin\theta_0 = q_2(2 - 2q_1)^{-1}$, on a cycle around an HQV core. Additionally, the presence of KLS walls results in winding of the $\hat{\mathbf{d}}$-vector by $2\theta_0$. These solitons can either extend between different pairs of HQVs, Fig. 4a, while walls with

total change $\Delta\theta = \pi$ are also possible if both solitons are located between the same pair of HQVs, Fig. 4b.

The appearance of KLS walls and the associated $\widehat{\mathbf{d}}$-solitons has the following consequences for NMR. The frequency shift of the bulk PdB phase in axial field for $q < 1/2$ is[45]

$$\Delta\omega_{\text{PdB},||} = \omega_{\text{PdB},||} - \omega_{\text{L}} \approx \left(1 + \frac{5}{2}q\right)\frac{\Omega_{\text{PdB}}^2}{2\omega_{\text{L}}}, \qquad (5)$$

where $\Omega_{\text{PdB}}$ is the Leggett frequency of the PdB phase, defined in the Supplementary Note 5. In transverse magnetic field, the bulk line has a positive frequency shift

$$\Delta\omega_{\text{PdB},\perp} = \omega_{\text{PdB},\perp} - \omega_{\text{L}} \approx (q - q^2)\frac{\Omega_{\text{PdB}}^2}{2\omega_{\text{L}}}, \qquad (6)$$

and winding of the $\widehat{\mathbf{d}}$-vector in a soliton leads to a characteristic frequency shift

$$\Delta\omega_{\text{PdBsat}} = \omega_{\text{PdBsat}} - \omega_{\text{L}} \approx \lambda_{\text{PdB}}\frac{\Omega_{\text{PdB}}^2}{2\omega_{\text{L}}}, \qquad (7)$$

where the dimensionless parameter $\lambda_{\text{PdB}}$ is characteristic to the defect. Numerical calculations in a 1D soliton model (Supplementary Note 3) for all possible solitons shown in Fig. 3a give the low-temperature values $\lambda_{\text{soliton}} \sim -0.8$ for $\pi - 2\theta_0$-soliton ("soliton") and $\lambda_{\text{big}} \sim -1.8$ for its antisoliton, which has $\pi + 2\theta_0$ winding ("big soliton"). The $2\theta_0$-soliton ("KLS soliton") related to the KLS walls outside spin-solitons gives rise to a frequency shift experimentally indistinguishable from the frequency shift of the bulk line. The last possibility, the "$\pi$-soliton" consisting of a KLS soliton and a soliton, c.f. Fig. 4b, gives $\lambda_\pi \sim -1.3$ at low temperatures. The measured value, $\lambda_{\text{PdB}} \sim -1.1$ at the lowest temperatures, as seen in Fig. 3a. The measured values for $\lambda_{\text{PdB}}$, together with the fact that the total winding of the $\widehat{\mathbf{d}}$-vector is also equal to $\pi$ in the PdA, and polar phases above the transition temperature suggest that the observed soliton structure in the PdB phase corresponds to the $\pi$-soliton in the presence of a KLS wall.

In addition, the KLS wall possesses a tension $\sim \xi q^3 \Delta_{\text{PdB}}^2 N_0$[32,33], where $N_0$ is the density of states. Thus, the presence of KLS walls applies a force pulling the two HQVs at its ends towards each other. The fact that the number of HQVs remains unchanged in the phase transition signifies that the KLS wall tension does not exceed the maximum pinning force in the studied nafen sample. This observation is in agreement with our estimation of relevant forces (Supplementary Note 4). Strong pinning of single-quantum vortices in B-like phase in silica aerogel has also been observed previously[46]. An alternative way to remove a KLS wall is to create a hole within it, bounded by a HQV[22]. Creation of such a hole, however, requires overcoming a large energy barrier related to creation of a HQV with hard core of the size of $\xi$. Moreover, growth of the HQV ring is prohibited by the strong pinning by the nafen strands. We also note that for larger values of $q$, there may exist a point at which the KLS wall becomes unstable towards creation of HQV pairs, and as a result, the HQV pairs bounded by KLS walls would eventually shrink to singly quantized vortices. For the discussion of the effect of nafen strands on the KLS walls, see Supplementary Note 4.

**Effect of rotation.** The density of HQVs created in the polar phase is controlled by the angular velocity $\Omega$ of the sample at the time of the phase transition from the normal phase, $n_{\text{HQV}} = 4\Omega\kappa^{-1}$, where $\kappa$ is the quantum of circulation. The integral of the NMR satellite depends on the total volume occupied by the solitons, whose width is approximately the spin-orbit length and the height is fixed by the sample size 4 mm. The

average soliton length is equal to the intervortex distance $\propto \Omega^{-1/2}$. Since the number of solitons is half of the number of HQVs, the satellite intensity scales as $\propto \Omega \times \Omega^{-1/2} = \sqrt{\Omega}$, which has been previously confirmed by measurements in the polar phase[20]. Here we observe similar scaling in the PdA and PdB phases, c.f. Fig. 3c.

Although the satellite intensity scales with the vortex density in the same way in both phases, there is one striking difference—the satellite intensity normalized to the total absorption integral in the PdB phase is smaller by a factor of $\sim 9$ relative to the PdA phase. Simultaneously, the original satellite intensity in the PdA phase is restored after a thermal cycle shown in Fig. 1b. Our numerical calculations of the soliton structure indicate that neither the PdB phase soliton width nor the oscillator strength would decrease substantially to explain the observed reduction in satellite size and the reason for the observed spectral intensity remains unclear—see Supplementary Note 7 for the calculations.

Another effect of rotation in the PdB phase transverse ($\mu = \pi/2$) NMR spectrum is observed at the main peak, c.f. Fig. 3b. The full-width-at-half-maximum (FWHM), extracted from the amplitude of the main peak assuming $w \times h = \text{const}$, where $w$ is its width and $h$ is height, scales as $\propto \sqrt{\Omega}$; Fig. 3d. Increase in the FWHM may indicate that the presence of KLS walls enhances scattering of spin waves and thus results in increased dissipation. Further analysis of this effect is beyond the scope of this article.

## Discussion

To summarize, we have found that HQVs, created in the polar phase of $^3$He in a nanostructured material called nafen, survive phase transitions to the PdA and PdB phases. Previously, HQVs have been reported in the polar phase[20], at the grain boundaries of $d$-wave cuprate superconductors[47], in chiral superconductor rings[48], and in Bose condensates[49,50]. Of these systems, only the polar phase contains vortex-core-bound fermion states as others are either Bose systems or lack the physical vortex core altogether. The domain walls with the sign change of a single gap component in $^3$He-B were suggested to interpret the experimental observations in bulk samples[51,52] ($q = 1$) and in the slab geometry[53]. Such walls, however, differ from those reported here as they are not bounded by strings but rather terminate at container walls. In the slab geometry, such walls are additionally topologically protected by a $\mathbb{Z}_2$ symmetry due to pinning of the $\widehat{\mathbf{l}}$ vector by the slab.

The survival of HQVs in the PdA and PdB phases has several important implications. First, HQVs in two-dimensional (2D) $p_x + ip_y$ topological superconductors (such as the A or PdA phases) are particularly interesting since their cores have been suggested to harbor non-Abelian Majorana modes, which can be utilized for topological quantum computation[54]. This fact has attracted considerable interest in practical realization of such states in various candidate systems[55–58]. While the PdA phase has the correct $p_x + ip_y$ type order parameter, scaling the sample down to effective 2D remains a challenge for future. However, the presence of the nafen strands, smaller in diameter than the coherence length, increases the separation of the zero-energy Majorana mode from other vortex-core-localized fermion states to a significant fraction of the superfluid energy gap, making it easier to reach relevant temperatures ($k_{\text{B}}T \lesssim$ energy separation of core-bound states) in experiments[59,60].

Second, we have shown how in the PdB phase, the HQVs, although topologically unstable as isolated defects, survive as composite defects known as "walls bounded by strings" (here KLS walls bounded by a pair of HQVs)—first discussed decades ago by Kibble, Lazarides, and Shafi in the context of cosmology[22]. Although the present existence of KLS walls in the context of the

Standard Model is shown to be unacceptable, as they either dominate the current energy density (first-order phase transition) or disappeared during the early evolution of the Universe (second-order phase transition), they occur in some GUTs and beyond-the-Standard-Model scenarios, especially in ones involving axion dark matter[61–63]. Any sign of similar defects in cosmological context would thus immediately limit the number of viable GUTs. Under our experimental conditions, the transition from the PdA phase to the PdB phase is weakly first-order ($q \ll 1$ at transition), but in principle, the order parameter allows a second-order phase transition to the PdB phase directly from the polar phase. Such a phase transition may be realized in future, e.g., by tuning confinement parameters. Studying the parameters affecting the amount of supercooling of the metastable PdA state ("false vacuum") before it collapses to the lowest-energy PdB state ("true vacuum") may also give insight on the nature of phase transitions in the evolution of the early Universe.

In conclusion, we have shown that the creation and stabilization of HQVs in different superfluid phases with controlled and tunable order-parameter structure is possible in the presence of strong pinning by the confinement. The survival of HQVs opens up a wide range of experimental and theoretical avenues ranging from non-Abelian statistics and topological quantum computing to studies of cosmology and GUT extensions of the Standard Model. Additionally, our results pave way for the study of a variety of further problems, such as different fermionic and bosonic excitations living in the HQV cores and within the KLS walls, and the interplay of topology and disorder provided by the confining matrix[64]. A fascinating prospect is to stabilize new topological objects in novel superfluid phases by tuning the confinement geometry[36–38], temperature, pressure, magnetic field, or scattering conditions[65].

## Methods

**Sample geometry and thermometry**. The $^3$He sample is confined within a 4-mm-long cylindrical container with ⌀4 mm inner diameter, made from Stycast 1266 epoxy; see Fig. 1a for illustration. The experimental volume is connected to another volume of bulk B phase, used for thermometry and coupling to nuclear demagnetization stage. This volume contains a commercial quartz tuning fork with 32 kHz resonance frequency, commonly used for thermometry in $^3$He[66,67]. The fork is calibrated close to $T_c$ against NMR signal from bulk $^3$He-B surrounding the nafen-filled volume. At lower temperatures, we use a self-calibration scheme[68] by determining the onset of the ballistic regime from the fork's behavior[69].

**Sample preparation**. To avoid paramagnetic solid $^3$He on the surfaces, the sample is preplated with ~2.5 atomic layers of $^4$He[65]. The HQVs are created by rotating the sample in zero magnetic field with angular velocity Ω while cooling the sample from the normal phase to the polar phase. Then the rotation is stopped since, based on our observations, the HQVs remain pinned (and no new HQVs are created) over all relevant time scales, at least for 2 weeks after stopping the rotation. The typical cooldown rate close to the critical temperature was of the order of 0.01 $T_c$ per hour to reduce the amount of vortices created by the Kibble–Zurek mechanism. Once the state had been prepared, the temperature was kept below the polar phase critical temperature until the end of the measurement.

**NMR spectroscopy**. Static magnetic field of 12–27 mT corresponding to NMR frequencies of 409–841 kHz is created using two coils oriented along and perpendicular to the axis of rotation. The magnetic field can be oriented at an arbitrary angle in the plane determined by the two main coils. Special gradient coils are used to minimize the field gradients along the directions of the main magnets. The magnetic field inhomogeneity along the rotation axis is $\Delta H_{ax}/H_{ax} \sim 10^{-4}$ and in the transverse direction an order of magnitude larger, $\Delta H_{tra}/H_{tra} \sim 10^{-3}$. The NMR pick-up coil, oriented perpendicular to both main magnets, is a part of a tuned tank circuit with quality factor $Q \sim 140$. Frequency tuning is provided by a switchable capacitance circuit, thermalized to the mixing chamber of the dilution refrigerator. We use a cold preamplifier, thermalized to a bath of liquid helium, to improve the signal-to-noise ratio in the measurements.

**Rotation**. The sample can be rotated about the vertical axis with angular velocities up to 3 rad s$^{-1}$, and cooled down to ~ 150 $\mu$K using ROTA nuclear demagnetization refrigerator. The refrigerator is well balanced and suspended against vibrational noise. The earth's magnetic field is compensated using two saddle-shaped coils installed around the refrigerator to avoid parasitic heating of the nuclear stage. In rotation, the total heat leak to the sample remains below 20 pW[67].

## Data availability

All the data supporting the findings are available from the corresponding author upon reasonable request.

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

## Acknowledgements

We thank V.V. Zavyalov and V.P. Mineev for useful discussions and related work on spin-solitons and HQVs. This work has been supported by the European Research Council (ERC) under the European Union's Horizon 2020 research and innovation programme (grant agreement no. 694248) and by the Academy of Finland (grant nos. 298451 and 318546). The work was carried out in the Low Temperature Laboratory, which is part of the OtaNano research infrastructure of Aalto University.

## Author contributions

The experiments were conducted by J.T.M. and J.R.; the sample was prepared by V.V.D. and A.N.Y.; the theoretical analysis was carried out by J.T.M., V.V.D., J.N., G.E.V., A.N.Y. and V.B.E.; numerical calculations were performed by J.N. and K.Z.; V.B.E. supervised the project; and the paper was written by J.T.M., J.N., G.E.V. and V.B.E., with contributions from all authors.
