## [Peer Review File · Nature Communications]

Reviewers' comments:

Reviewer #1 (Remarks to the Author):

The group of the authors in the present manuscript has previously succeeded to observe the long-sought topological objects, half-quantum vortices (HQVs), in the polar phase of the superfluid ^3He realized in the strongly anisotropic aerogels including Nafen. The present manuscript corresponds to a subsequent report of a new observation of essentially the same group (Ref.1), and the authors report their observations of realization of HQVs in the polar-distorted A (PdA) and PdB phases lying at lower temperatures than the polar phase. In particular, the appearance of HQVs in the PdB phase is quite unexpected from the viewpoint of the topological classification. The authors identify them with the unavoidable defects formed via the Kibble-Zurek mechanism and try to analyze their NMR data thoroughly based on a phenomenological model. The observation of HQVs in PdB phase is fascinating, and their explanation on NMR data is clear.

Nevertheless, we have the following questions:

1) In contrast, it is felt that their explanation on the appearance of HQVs in PdA phase, given above eq.(3), is too simple. For instance, it should be clarified whether the authors believe that the HQV pair connected by the spin soliton is not stabilized for a larger value of $|b|$.

2) Through the final sentence in the paragraph {Topologically stable ...} of page 2, it is felt that the authors have argued a continuous change of the present HQV pair stabilized via the K-Z wall to the core of the well-known nonaxisymmetric vortex stabilized via the planar soliton. But they do not seem to be continuously transformed to each other. This point should be clarified.

Reviewer #2 (Remarks to the Author):

The observation of half-quantum vortices in the polar phase of topological superfluid ^3He was a landmark result from the same group, recently published in Phys. Rev. Lett 117, 255301 (2016). This work was in turn enabled by the discovery (confirming theoretical prediction) that the polar phase could be stabilised in superfluid ^3He in nematically ordered aerogel, of sufficiently high density (nafen-90) Phys. Rev. Lett. 115, 165304 (2015). This latter paper also showed that at lower temperatures transitions occur first to polar distorted A phase (PdA), then to polar distorted B-Phase (PdB).

This manuscript reports that HQVs survive in the PdA phase, and also survive thermal cycling into PdB and back to PdA. Furthermore it claims to observe, through their NMR response, defects in PdB that are identified as composite defects, in which HQVs are bound to a Kibble-Lazarides-Shafi (KLS) domain wall, originally proposed in the context of symmetry breaking transitions in the early universe. There are a variety of KLS walls possible; the NMR response of each is calculated and compared to experiment. Thus the experiment not only identifies a KLS wall for the first time, but points to which of the possibilities is found in the experiment.

In all cases the density of these topological defects can be controlled via sample rotation on the initial cool-down into the superfluid phases. A residual density of defects is attributed to the Kibble-Zurek mechanism on initial cooling through T_c , and dependent on the rate of cooling (although that is not explicitly reported in current manuscript).

These results are novel and of wide interest, meriting publication in Nature Communications. They significantly extend the earlier reported work. The observation of composite defects (in PdB) is highly

relevant to the further development of superfluid ^3He as a model laboratory system for cosmology, with potentially high impact on the understanding of phase transitions in the early universe. The observation of HQVs in PdA, mentioned in a preliminary way in Phys. Rev. Lett 117, 255301 (2016) is relevant to the development of topological quantum computation: stable HQVs in a chiral superconductor host Majorana fermions at their core (but see also below).

There now follow some more detailed comments/criticisms.

1. This paper is authored by experimentalists and theorists. More care is needed throughout to distinguish between statements about experimental findings, and theoretical predictions. There are some cases where, on the basis of the evidence, the former should be more qualified and less definitive. On p2, column 2, paragraph 2 “Topologically stable HQVs...”; each statement is presented as fact. Actually they are the conclusions drawn from the experiment described, with varying degrees of certainty.

2. The introduction to the paper is excellent and stresses the analogues between superfluid ^3He and the universe. However, it is also necessary to highlight distinctions. Aerogel impurities were first introduced into superfluid ^3He in the mid-90s as a means to study the effect of disorder on a p-wave paired superfluid. The early universe was not like that, as far as I know. Does it matter? Nafen-90 is described in Phys. Rev. Lett 117, 255301 (2016) as a “nematically ordered aerogel-like material”, which is accurate. Here it is described as a “nanostructured nematically ordered material”, which is potentially misleading (due to the randomness element in strand separation and orientation). Furthermore the strong pinning of HQVs by the aerogel strands needs to be given far more emphasis in the opening introduction [it is well treated in Phys. Rev. Lett 117, 255301 (2016)]. This may weaken cosmological analogue, but gives a balanced picture. Pinning is referred to in the final paragraph, but it needs to be more explicitly discussed in the introduction.

3. The evidence for HQVs in PdA is persuasive, but not totally definitive. It seems to be based on relatively little data, and the dependence of frequency shift on field angle, to compare with that previously observed in the polar phase, is absent. Data on the $\sqrt{\Omega}$ (rotation speed) dependence of satellite intensity in PdA is included in Fig 3, title of which is “HQVs in the PdB phase”. Some redrafting might help to better convey message, with an explicit acknowledgement of what more could be done. Despite these caveats, this is an important and exciting result.

4. Given that evidence is presented for HQVs in a chiral superfluid (PdA), and the importance of that, I recommend that a full discussion of the vortex core is included. This object is more like a vortex than a fluxoid state [ref 45]. A nice discussion is at the end of Phys. Rev. Lett 117, 255301 (2016). I recommend that a version of this (updated) is included. The discussion is worthy of some repetition in this context. As I understand it, the point is that the diameter of nafen strand pinning the HQV is smaller than the coherence length, that this modifies the energy levels of the core excitation levels to increase the gap from the zero energy Majorana state (if indeed that survives?).

5. There is little if any discussion on the factors influencing the supercooling of PdA. This is an important issue as, in addition to intrinsic interest, it relates to the cosmological question of phase transitions in the early universe, which the manuscript discusses at length in the introduction. See ref 1 of SI, Nature Comms. 8, 15963 (2016), and refs therein. This issue is at least worthy of mention, and if the authors have any data, they might consider adding it.

6. The role of the disorder, and scale of confinement can be changed by using lower density aerogel. In such aerogels PdA and PdB are identified from NMR studies by Dmitriev group, see refs in Phys. Rev. Lett. 115, 165304 (2015), but no polar phase. However torsional oscillator experiments, Nature Comms. 7, 12975 (2016), which are arguably more sensitive to presence of polar phase, seem to detect it. In the context of the present manuscript, this opens the possibility of studying the new topological defects in lower density aerogel. I believe it would be helpful to mention this work and these issues, since the details of nematic aerogel structure may influence results and their interpretation, and this should be discussed.

7. The description of the KLS domain walls in the PdB phase is clear, at a good level, with appropriate supporting material in SI. There are many nanofibrillar strands which intercept a KLS domain wall. These strands pin the HQVs. What do they do to the domain wall?

8. It would be helpful to a wider readership, to refer to the future prospects of work on “properly engineered nanostructured confinement”, to quote Phys. Rev. Lett 117, 255301 (2016). In the last sentence of the present manuscript, tuning by confinement (geometry) is also mentioned. Two papers should be referred to in the main text: Phys. Rev. B92, 144515 (2015), Superfluid phases of ^3He in nanoscale channels (theory); Science 340, 841 (2013) Phase diagram of topological superfluid ^3He confined in a nanoscale slab geometry (experiment). I submit that they are more relevant than the current ref 60. This will give appropriate attention to the real achievements in engineering the relevant geometries, measuring superfluid ^3He in them, and the methods for predicting the phase diagram. This approach constitutes a significant development, away from anisotropic aerogels as porous media, with their lack of precise control of “confinement” and their disorder, which should be brought to the attention of the readers of this Nature Communication.

I hope that the above comments are taken into account in a revised version of the manuscript. They are advisory.

As stated previously the manuscript presents novel results and ideas of wide interest, meriting publication in Nature Communications.

Reviewer #1

1) In contrast [to explanation of HQVs in the PdB phase], it is felt that their explanation on the appearance of HQVs in PdA phase, given above eq.(3), is too simple. For instance, it should be clarified whether the authors believe that the HQV pair connected by the spin soliton is not stabilized for a larger value of $|b|$.

We have extended the discussion of HQV stability in the PdA phase above Eq. (3). Indeed, with increasing $|b|$ the soliton tension increases. However, even for $|b| = 1$ and without pinning, a pair of HQVs, once created, will not collapse, since repulsion between two vortices with the same circulation will cancel the tension from the soliton at the equilibrium distance significantly exceeding the core size.

2) Through the final sentence in the paragraph Topologically stable ... of page 2, it is felt that the authors have argued a continuous change of the present HQV pair stabilized via the K-Z wall to the core of the well-known nonaxisymmetric vortex stabilized via the planar soliton. But they do not seem to be continuously transformed to each other. This point should be clarified.

Two HQVs in the PdB phase are connected by the KLS wall, which has (distorted) planar phase in its core (in our notation, $q_2 = 0, |q_1| > 0$). Thus, we believe that continuous transition to a singly quantized vortex with nonaxisymmetric core connected with (distorted) planar soliton (as formulated by the reviewer) is, in principle, possible. Theoretical analysis of this scenario in $^3\text{He-B}$ can be found from supplementary references [13-15]. Detailed analysis for the PdB phase in the presence of disorder and pinning is, however, beyond the scope of this manuscript. We have relaxed the wording in this sentence to admit that the scenario is, in the absence of experimental proof, based on our theoretical understanding.

Reviewer #2

1. This paper is authored by experimentalists and theorists. More care is needed throughout to distinguish between statements about experimental findings, and theoretical predictions. There are some cases where, on the basis of the evidence, the former should be more qualified and less definitive. On p2, column 2, paragraph 2 Topologically stable HQVs; each statement is presented as fact. Actually they are the conclusions drawn from the experiment described, with varying degrees of certainty.

The wording throughout the mentioned paragraph was changed to more accurately reflect which statements are based on theoretical understanding, and which are based on experimental find-

ings. We have made similar modification in other parts of the text where theoretical understanding and experimental results are discussed together.

2. The introduction to the paper is excellent and stresses the analogues between superfluid ^3He and the universe. However, it is also necessary to highlight distinctions. Aerogel impurities were first introduced into superfluid ^3He in the mid-90s as a means to study the effect of disorder on a p -wave paired superfluid. The early universe was not like that, as far as I know. Does it matter?

In our scenario, the confinement is primarily used to modify the order parameter. We believe that the underlying physics of formation and combining of topological defects at symmetry-breaking phase transitions remains unaffected by impurities. The reason is that the relevant length scale (thickness of the domain wall) exceeds the distance between nafen strands by more than an order of magnitude. We have significantly enhanced the discussion of length scales in particular by adding a new Supplementary note 5 on the role of pinning.

Nafen-90 is described in Phys. Rev. Lett 117, 255301 (2016) as a nematically ordered aerogel-like material, which is accurate. Here it is described as a nanostructured nematically ordered material, which is potentially misleading (due to the randomness element in strand separation and orientation).

Description of nafen has been changed to “nematically ordered aerogel-like material” in the introduction.

Furthermore the strong pinning of HQVs by the aerogel strands needs to be given far more emphasis in the opening introduction [it is well treated in Phys. Rev. Lett 117, 255301 (2016)]. This may weaken cosmological analogue, but gives a balanced picture. Pinning is referred to in the final paragraph, but it needs to be more explicitly discussed in the introduction.

Pinning is indeed addressed in the last paragraph of introduction. The role of pinning in our experiments – namely allowing studies of the initial frozen out-of-equilibrium state but preventing studies of vortex dynamics, is now highlighted. Note that the introduction has to obey the Nature Communications word limit. More discussion is present in a new Supplementary note 5.

3. The evidence for HQVs in PdA is persuasive, but not totally definitive. It seems to be based on relatively little data, and the dependence of frequency shift on field angle, to compare with that previously observed in the polar phase, is absent. Data on the $\sqrt{\Omega}$ (rotation speed) dependence of satellite intensity in PdA is included in Fig 3, title of which is HQVs in the PdB phase. Some redrafting might help to better convey message, with an explicit acknowledgement of what more could be done. Despite these caveats, this is an important and exciting result.

We have not performed complete field-direction-dependence measurements for the satellite frequency shift in the spirit of PRL 117, 255301 (2016), but rather measured 3 angles: $\mu = 0$, 40.5 and 90 degrees. The measurements confirm the expected angular dependence with approximately temperature-independent $\lambda \approx -0.9$. The data at $\mu = 90^\circ$ show the largest absolute frequency shift and cover the widest temperature range, so they are used in Fig. 3. This value of λ coincides with our previous measurements in the polar phase (although, as discussed in the text, the absolute frequency shifts differ due to the different Leggett frequency) and, as such, we consider the amount of evidence for the HQVs to be sufficient. Nevertheless, we added a sentence to the end of the PdA section to suggest that a more detailed study on the field angle dependence could be done in the future.

4. *Given that evidence is presented for HQVs in a chiral superfluid (PdA), and the importance of that, I recommend that a full discussion of the vortex core is included. This object is more like a vortex than a fluxoid state [ref 45]. A nice discussion is at the end of Phys. Rev. Lett 117, 255301 (2016). I recommend that a version of this (updated) is included. The discussion is worthy of some repetition in this context. As I understand it, the point is that the diameter of nafen strand pinning the HQV is smaller than the coherence length, that this modifies the energy levels of the core excitation levels to increase the gap from the zero energy Majorana state (if indeed that survives?).*

Indeed, this point was addressed in Phys. Rev. Lett 117, 255301 (2016). By reviewer recommendation we have included a brief summary, with relevant references, to the end of the second paragraph in “Discussion”.

5. *There is little if any discussion on the factors influencing the supercooling of PdA. This is an important issue as, in addition to intrinsic interest, it relates to the cosmological question of phase transitions in the early universe, which the manuscript discusses at length in the introduction. See ref 1 of SI, Nature Comms. 8, 15963 (2016), and refs therein. This issue is at least worthy of mention, and if the authors have any data, they might consider adding it.*

Within the accuracy of our measurements of temperature, the supercooled transition from the PdA to the PdB phase occurs at a stable value $T \approx 0.57T_c$, with very little scatter between measurements. The value does not seem to depend e.g. on the history (whether or not this was the first cooldown to the PdB phase) or other parameters in our measurements, although we have no conclusive data and a detailed investigation on the topic remains a task for the future. However, we revisited our temperature calibration and updated temperatures in Figs. 2, 3 (Fig. + caption), and Supplementary Figure 5. The temperatures for the supercooled transition and the transition on heating ($T \approx 0.78T_c$) are now marked in Fig. 3 (a). We have also now commented on thermometry with relevant references in the “Methods” section, where the paragraph “Sample geometry” is now “Sample geometry and thermometry”. As for the origin of this particular value for the supercooled transition temperature, we could not make any conclusions as we have not explored supercooling as a function of various parameters. This point is now addressed in the last sentence of the third paragraph of “Discussion”.

6. *The role of the disorder, and scale of confinement can be changed by using lower density aerogel. In such aerogels PdA and PdB are identified from NMR studies by Dmitriev group, see refs in Phys. Rev. Lett. 115, 165304 (2015), but no polar phase. However torsional oscillator experiments, Nature Comms. 7, 12975 (2016), which are arguably more sensitive to presence of polar phase, seem to detect it. In the context of the present manuscript, this opens the possibility of studying the new topological defects in lower density aerogel. I believe it would be helpful to mention this work and these issues, since the details of nematic aerogel structure may influence results and their interpretation, and this should be discussed.*

The possibilities to stabilize new topological defects (potentially in new superfluid phases) by tuning the confinement are indeed fascinating. We address this question now in the last paragraph of the discussion, where we have added this reference along those mentioned in comment 8.

7. *The description of the KLS domain walls in the PdB phase is clear, at a good level, with appropriate supporting material in SI. There are many nafen strands which intercept a KLS domain wall. These strands pin the HQVs. What do they do to the domain wall?*

This is an important question. We have added a Supplementary Note 5 “Pinning of HQV by a columnar defect” focusing to this problem. This note is referred to in the last sentence of the “Half-quantum vortices in PdB phase” section. For clarity of discussion we have introduced a new parameter, “wall thickness” $\xi_W = q^{-1}\xi$, which is used now in the text. We have also corrected a misprint in wall tension formula at page 5, column 2, second paragraph ($q^2 \rightarrow q^3$).

8. *It would be helpful to a wider readership, to refer to the future prospects of work on properly engineered nanostructured confinement, to quote Phys. Rev. Lett 117, 255301 (2016). In the last sentence of the present manuscript, tuning by confinement (geometry) is also mentioned. Two papers should be referred to in the main text: Phys. Rev. B92, 144515 (2015), Superfluid phases of 3He in nanoscale channels (theory); Science 340, 841 (2013) Phase diagram of topological superfluid 3He confined in a nanoscale slab geometry (experiment). I submit that they are more relevant than the current ref 60. This will give appropriate attention to the real achievements in engineering the relevant geometries, measuring superfluid 3He in them, and the methods for predicting the phase diagram. This approach constitutes a significant development, away from anisotropic aerogels as porous media, with their lack of precise control of confinement and their disorder, which should be brought to the attention of the readers of this Nature Communication.*

References to the proposed publications were added to the last sentence of Discussion.

Further changes

- Rewrote the abstract to satisfy the 150 word limit.
- Reformulated some of the Introduction to satisfy the 1000 word limit.
- Changed all references to SI to comply with the guidelines.
- Changed colors in Figs. 2 and 3 to avoid combination of red+green.
- Removed the reference to the ROTA demagnetization cryostat from methods. section “Rotation” to satisfy the reference limit.
- Made a few other small modifications throughout the text to enhance readability.

REVIEWERS' COMMENTS:

Reviewer #2 (Remarks to the Author):

The authors have, for the most part, carefully considered the comments of the reviewers and revised the manuscript accordingly.

In some cases additional material has been included in the Supplementary Information.

Just one point. I am still not convinced by the response to reviewer 2 point 6. This has to do with the presence or not of polar phase near T_c in lower density aerogels, and whether the more open aerogel samples support HQVs in PdA. A sentence has been added on p3, right column, line 4: "Indeed, in confined geometry, where PdA is observed immediately below T_c , no HQVs are found [ref 44]". I don't see how that can be accurate as ref 44 does not test for HQVs. It is quite hard, for me at least, to track a clear statement summarising the survival of HQVs in nafen samples of different density, across the various papers, including ref. 21.

If the manuscript could be tweaked accordingly that would be good.

The above could be construed as a minor detail.

[I do not list minor errors: eg p2 third line from bottom, exists should be exist.]

As a default I recommend publication of the manuscript in its current form, without delay.

Reviewer #2

Just one point. I am still not convinced by the response to reviewer 2 point 6. This has to do with the presence or not of polar phase near T_c in lower density aerogels, and whether the more open aerogel samples support HQVs in PdA. A sentence has been added on p3, right column, line 4: "Indeed, in confined geometry, where PdA is observed immediately below T_c , no HQVs are found [ref 44]". I don't see how that can be accurate as ref 44 does not test for HQVs. It is quite hard, for me at least, to track a clear statement summarising the survival of HQVs in nafen samples of different density, across the various papers, including ref. 21.

The referee is correct that the reference [44] does not contain statements about creation of HQVs in the nafen sample with (Pd)A phase appearing immediately below T_c . In fact, our preliminary measurements show that HQVs are not formed in the transition from the normal to PdA, at least if the PdA phase is closer to the A phase than to the polar phase (large b). Again, these results are preliminary and not systematic – therefore we reformulated the sentence to “Studying whether HQVs are formed in the transition from the normal phase to the PdA phase with finite polar distortion ($0 < b < 1$) remains a task for the future.” We also added the word “undistorted” to the previous sentence to emphasize that we discuss the pure (undistorted) A phase.

Further changes

- We reformulated the first sentence of the first paragraph on page 2 “Here we focus on..” → “Of particular interest are...” as suggested by the Editor. This removes the misleading impression that we discuss the results of the current Article in this paragraph.
- Inset of Fig. 3 (b) now contains graphical illustration of the uncertainty in the frequency space - converted to λ and denoted by error bars in the (a) panel. Description of the error bars in the caption is also clarified. We stress that the error bars are not of statistical origin: there is no averaging of the data here. The error bars are simply indicators of the spectral line width.